# Pro-Apoptotic Activity of MCL-1 Inhibitor in Trametinib-Resistant Melanoma Cells Depends on Their Phenotypes and Is Modulated by Reversible Alterations Induced by Trametinib Withdrawal

**DOI:** 10.3390/cancers15194799

**Published:** 2023-09-29

**Authors:** Mariusz L. Hartman, Paulina Koziej, Katarzyna Kluszczyńska, Małgorzata Czyz

**Affiliations:** Department of Molecular Biology of Cancer, Medical University of Lodz, 92-215 Lodz, Poland; mariusz.hartman@umed.lodz.pl (M.L.H.); paulina.tecza1@stud.umed.lodz.pl (P.K.); katarzyna.kluszczynska@umed.lodz.pl (K.K.)

**Keywords:** cancer cell plasticity, drug holiday, drug rechallenge, MCL-1 inhibitor, melanoma, S63845, trametinib resistance

## Abstract

**Simple Summary:**

Skin melanoma is fully curable by surgery if diagnosed at an early stage. Targeted drugs against BRAF^V600^ and MEK1/2 were the first therapeutic breakthrough for patients with advanced melanoma. However, acquired drug resistance resulting in disease relapse interrupts the treatment after a few months. Therefore, new treatment strategies are imperative to address this problem. The multifactorial nature of melanoma drug resistance involves genetic alterations but also an extraordinary capability of melanoma to adapt to changes in the microenvironment including drug holiday and rechallenge. The alterations in the levels of proteins impacting the MCL-1 activity and stability, such as p-ERK1/2, BIM and NOXA, could explain the differences in pro-apoptotic response to MCL-1 inhibitors between trametinib-resistant melanoma cells grown with and without trametinib. Our results underline the importance of a more detailed analysis of resistant melanoma phenotypes in preclinical studies and in stratifying relapsed patients for clinical trials evaluating novel treatment strategies.

**Abstract:**

Background: Although BRAF^V600^/MEK inhibitors improved the treatment of melanoma patients, resistance is acquired almost inevitably. Methods: Trametinib withdrawal/rechallenge and MCL-1 inhibition in trametinib-resistance models displaying distinct p-ERK1/2 levels were investigated. Results: Trametinib withdrawal/rechallenge caused reversible changes in ERK1/2 activity impacting the balance between pro-survival and pro-apoptotic proteins. Reversible alterations were found in MCL-1 levels and MCL-1 inhibitors, BIM and NOXA. Taking advantage of melanoma cell dependency on MCL-1 for survival, we used S63845. While it was designed to inhibit MCL-1 activity, we showed that it also significantly reduced NOXA levels. S63845-induced apoptosis was detected as the enhancement of Annexin V-positivity, caspase-3/7 activation and histone H2AX phosphorylation. Percentages of Annexin V-positive cells were increased most efficiently in trametinib-resistant melanoma cells displaying the p-ERK1/2^low^/MCL-1^low^/BIM^high^/NOXA^low^ phenotype with EC_50_ values at concentrations as low as 0.1 μM. Higher ERK1/2 activity associated with increased MCL-1 level and reduced BIM level limited pro-apoptotic activity of S63845 further influenced by a NOXA level. Conclusions: Our study supports the notion that the efficiency of an agent designed to target a single protein can largely depend on the phenotype of cancer cells. Thus, it is important to define appropriate phenotype determinants to stratify the patients for the novel therapy.

## 1. Introduction

The personalized treatment of cancer has predominantly focused on genetic alterations. The association between genetic variants and treatment response has led to the development of therapies targeting pathways hyper-activated in cancer such as the BRAF^V600^/MEK/ERK pathway in melanoma [1,2,3,4,5]. While targeted therapies against B-Raf proto-oncogene (BRAF^V600^) and its downstream target mitogen-activated protein kinase kinase 1/2 (MEK1/2) are beneficial for patients with *BRAF* mutant unresectable melanoma, acquired resistance remains a fundamental clinical challenge [1,2,3,4,5]. Immunotherapy with checkpoint inhibitors, if not previously given, is currently the treatment of choice for relapsed melanoma patients, however, almost 50% of patients do not respond, and most responders eventually progress [6,7]. Other options include available clinical trials, treatment beyond progression, re-exposure to targeted therapy after drug holiday or another treatment [8,9]. Results of clinical studies on the efficacy and safety of rechallenge with BRAF^V600^/MEK inhibitors are limited and conflicting [8,10,11,12,13,14,15,16]. Various mechanisms of resistance to BRAF^V600^/MEK inhibitors have been identified, and novel treatment modalities for relapsed patients are highly needed [17,18,19,20,21,22,23,24,25]. Therefore, a better understanding of mechanisms of resistance to targeted therapy is needed to identify melanoma patients who either may benefit from the rechallenge therapy or may require other treatment. The majority of the studies focus on an early response of melanoma cells to targeted therapeutics manifested as the selection of the subpopulation of drug-tolerant persister cells [26,27,28,29], which may provide clinical benefits by preventing the development of resistance. However, finding novel treatment modalities for melanoma patients who already developed resistance to targeted therapies is a demanding task because of melanoma genomic heterogeneity and notable phenotypic plasticity of resistant melanoma cells [17,21,22,30,31,32,33]. In contrast to studies of early response to targeted therapies, useful preclinical melanoma models of stable resistance are limited. We have previously characterized diverse models of stable resistance to either BRAF^V600^ inhibitor vemurafenib or MEK1/2 inhibitor trametinib obtained from patient-derived drug-naïve melanoma cell lines by long-term treatment with increasing concentrations of drugs [22]. Among eleven melanoma cell lines investigated in that study, there was no similar pattern of genetic and non-genetic changes induced during the development of resistance [22]. For the present study, we have chosen two BRAF^V600^ melanoma cell lines that differed in the activity of the BRAF/MEK/ERK pathway after they developed resistance to trametinib (TRA), 29_TRAR (p-MEK1/2^low^/p-ERK1/2^low^) and 21_TRAR (p-MEK1/2^high^/p-ERK1/2^high^). The resistance to trametinib has been chosen as this drug is broadly used in combined treatment modalities and clinical trials for melanoma but also other cancers [34,35,36,37,38,39,40]. We aimed to decipher the adaptive alterations in trametinib-resistant cells associated with alternating periods of drug withdrawal and rechallenge and find the new vulnerability that could be potentially targeted. Our study revealed that in addition to reversible changes in the activity of BRAF/MEK/ERK signaling, drug holiday/rechallenge influenced the balance between pro-survival protein MCL-1 and its endogenous inhibitors BIM and NOXA. This in turn affected the activity of S63845, one of several synthetic inhibitors of the MCL-1 protein recently developed and investigated in preclinical studies [41]. Abnormally enhanced expression of pro-survival proteins and reduced expression of pro-apoptotic proteins can help cancer cells to evade apoptosis and largely contribute to the development of resistance to therapies, but growing evidence also indicates that these cancer cell- and/or drug-driven disturbances might also create apoptotic vulnerabilities that can be determined and potentially therapeutically exploited (recently reviewed in [42]). Moreover, in view of our results showing that p-ERK1/2^high^ and p-ERK1/2^low^ trametinib-resistant cell lines differed in response to MCL-1 inhibitor, a broad analysis of the main molecular determinants of activity of a novel therapeutic target seems to be necessary and should be preferentially performed on resistant cells in the presence of a drug that caused resistance and after drug cessation to define the requirements for a highly efficacious new treatment regimen.

## 2. Materials and Methods

### 2.1. Compounds

Trametinib from Selleck Chemicals LLC (Houston, TX, USA) was used at 50 nM. BH3-mimetics such as S63845, ABT-199, and ABT-263 (Selleck Chemicals LLC, Houston, TX, USA) were used at 1 µM unless otherwise indicated.

### 2.2. Cell Lines and Cultures

Drug-naïve melanoma cell lines were obtained from tumor specimens [43]. The study was approved by the Ethical Commission of the Medical University of Lodz (approval number: RNN/84/09/KE), and informed consent was obtained from each patient. Trametinib-resistant cell lines (21_TRAR and 29_TRAR) were obtained as described previously [44]. Cell lines were cultured with or without trametinib in serum-free stem cell medium (SCM) in low-adherent flasks at 37 °C in a humidified atmosphere containing 5% CO_2_ [44]. The medium was exchanged twice a week. For “drug holiday” experiments, cells were grown in a medium without trametinib for up to 22 days (DH 1-22 d and re-DH 1-22 d). During the rechallenge with the drug, trametinib at 50 nM was reintroduced to the culture medium (re-TRA 1-22 d). A LookOut Mycoplasma qPCR Detection kit (Sigma-Aldrich, St. Louis, MO, USA) was utilized to test the potential contamination of cell cultures, and the results were negative.

### 2.3. Annexin V/Propidium Iodide (PI) Staining and Flow Cytometry

Melanoma cells were exposed to S63845, ABT-199, or ABT-263 for 44 h. Cells were collected, trypsinized, and double-stained with an Apoptosis detection kit consisting of PI and FITC-conjugated Annexin V (BD Biosciences, San Jose, CA, USA). Samples were measured by flow cytometer FACSVerse (BD Biosciences) and analyzed with the BD FACSuite software v1.0.5.3824.

### 2.4. Acid Phosphatase Activity (APA) Assay

An acid phosphatase activity (APA) assay was used to assess changes in overall cell viability. Melanoma cells were plated in 96-well plates at a density of 3.5 × 10^3^ viable cells per well. Then, they were exposed to BH3-mimetics: S63845, ABT-263, or ABT-199 at 1 µM for 24, 48, and 72 h. At each time point, the plates were centrifuged, and the culture medium was replaced with 100 μL of assay buffer consisting of 0.1 M sodium acetate (pH = 5), 5 mM p-nitrophenyl phosphate (Merck Life Science, Mississauga, ON, Canada), and 0.1% Triton X-100 (Merck Life Science, Burlington, MA, USA). The plates were incubated at 37 °C for 2 h. Ten (10) μL of 1 M NaOH was added to each well to stop the reaction. Absorbance was measured at 405 nm using a microplate reader Infinite M200Pro (Tecan Group Ltd., Salzburg, Austria).

### 2.5. Caspase Activation by Time-Lapse Fluorescence Microscopy (IncuCyte ZOOM)

Melanoma cells were seeded in 96-well plates (8 × 10^3^ cells per well) and exposed to 1 µM S63845, ABT-199, or ABT-263. IncuCyte™ Caspase-3/7 Apoptosis Assay Reagent (Sartorius, Göttingen, Germany) was added at 4 μM to each well. Activation of caspase-3/7 was monitored every 4 h using a time-lapse fluorescence microscope system (IncuCyte, Essen Bioscience, Essen, Germany). Quantification of the images was performed with the IncuCyte^®^ ZOOM basic analyzer. The percentages of cells with active caspase-3/7 were calculated as the percentage of the confluence of apoptotic cells divided by the percentage of the confluence of all cells.

### 2.6. Cell Lysate Preparation and Western Blotting

RIPA buffer (50 mM Tris-HCl pH = 8.0, 150 mM NaCl, 1% Triton X-100, 0.5% sodium deoxycholate, 0.1% SDS) was used to lyse melanoma cells. A cocktail of protease and phosphatase inhibitors was freshly added (Sigma-Aldrich). After 30 min, samples were centrifuged (17,968× *g*, 15 min, 4 °C), cell lysates were collected and the protein concentration was determined using the Bradford assay (BioRad, Hercules, CA, USA) at 595 nm. The cell lysates were diluted in 2× Laemmli buffer (125 mM Tris-HCl pH = 6.8, 0.004% bromophenol blue, 20% glycerol, 4% SDS, and 10% β-mercaptoethanol). Protein samples (15 μg) were loaded on either 7% or 12% SDS-polyacrylamide gel, and electrophoresis was run at a constant voltage of 25 V/cm. The proteins were transferred onto Immobilon-P and Immobilon-PSQ (Merck Millipore, Billerica, MA, USA) from 7% and 12% SDS-polyacrylamide gel, respectively. Either 5% non-fat milk or phosphoBLOCKER (Cell Biolabs, San Diego, CA, USA) in PBS containing 0.05% Tween-20 (Sigma-Aldrich) was used to block the membranes for 45 min. Primary antibodies against BIM (#2933), NOXA (#14766), MCL-1 (#94296), phospho-histone H2A.X (Ser139; #2577), phospho-MEK1/2 (Ser217/221; #9154), phospho-ERK1/2 (Thr202/Tyr204; #4377), MEK1/2 (#4694), ERK1/2 (#9107) (Cell Signaling, Danvers, MA, USA), and GAPDH (sc-47724) (Santa Cruz Biotechnology, Santa Cruz, CA, USA) were used overnight at dilution 1:1000. The membranes were subsequently incubated with secondary anti-rabbit (#7074) or anti-mouse (#7076) HRP-linked antibodies (Cell Signaling) used at 1:5000 for 1 h at room temperature. After washing, the Clarity™ Western ECL Substrate (Bio-Rad) was added for 1 min and chemiluminescence was visualized using a ChemiDoc Imaging System (Bio-Rad). For quantification, freely accessible ImageJ software was used.

### 2.7. RNA Isolation, cDNA Synthesis, and Quantitative Real-Time PCR (qRT-PCR)

Total RNA Mini kit (A&A Biotechnology, Gdynia, Poland) was used to extract total RNA. The procedure was performed according to the manufacturer’s protocol. RNA concentration and purity were assessed using a microplate reader Infinite M200Pro (Tecan) at 260 nm and with a 260/280 nm ratio, respectively. Complementary DNA (cDNA) was synthesized using 1000 ng of total RNA, 300 ng of random primers, and 1 μL of SuperScript^®^ II Reverse Transcriptase (Invitrogen Thermo Fisher Scientific, Carlsbad, CA, USA). To assess transcript levels of target genes, a quantitative real-time polymerase chain reaction was performed using the Rotor-Gene 3000 Real-Time DNA analysis system (Corbett Research, Mortlake, VI, Australia). cDNA was amplified using 200 nM forward primer, 200 nM reverse primer, KAPA SYBR FAST qPCR Kit Universal 2X qPCR Master Mix (Sigma-Aldrich), and 25 ng of cDNA with annealing temperature 56 °C. The sequences of forward and reverse primers for NOXA and MCL-1 were shown elsewhere [45,46]. To calculate the relative level of each transcript versus a reference gene RPS17 (primers: 5′-AAT CTC CTG ATC CAA GGC TG-3′ and 5′-CAA GAT AGC AGG TTA TGT CAC G-3′), a mathematical model with an efficiency correction was used [47]. (Appendix A)

### 2.8. Whole-Exome Sequencing (WES) and WES Data Analysis

The procedures of extraction of DNA, whole-exome sequencing, and analysis of Whole-Exome Sequencing (WES) data were described previously [22,48]. Raw WES data for drug-naïve cell lines, DMBC21 and DMBC29, are accessible at ArrayExpress (E-MTAB-6978) and European Nucleotide Archive (ERP109743). Raw WES data for trametinib-resistant cell lines (21_TRAR and 29_TRAR) are available under the numbers E-MTAB-7248 at ArrayExpress and ERP111109 at European Nucleotide Archive. Freely accessible Polyphen-2 software (http://genetics.bwh.harvard.edu/pph2/ URL accessed on 21 August 2023) was used to predict the functional effects of single nucleotide polymorphisms (SNPs) in genes encoding components of the core apoptotic machinery. The Polyphen-2-based prediction for each variant was classified as probably damaging (score range: 0.960–1.000), possibly damaging (score range: 0.450–0.959), or benign (score range: 0.000–0.449).

### 2.9. Statistical Analysis

Results originating from at least three independent experiments unless otherwise indicated are presented as mean values ± standard deviation (S.D.). The statistical significance was determined by a two-tailed unpaired Student’s *t*-test. *p* ≤ 0.05 (*), *p* ≤ 0.01 (**), *p* ≤ 0.001 (***).

## 3. Results

### 3.1. Adaptive Alterations Induced by Drug Withdrawal and Re-Exposure to Trametinib Are More Pronounced in Trametinib-Resistant Melanoma Cells Displaying Low ERK1/2 Activity Than Those with High ERK1/2 Activity

To inquire phenotypic plasticity of trametinib-resistant melanoma cells, changes in activities of crucial components of BRAF/MEK/ERK signaling induced during alternating periods of trametinib withdrawal and rechallenge were investigated for their reversibility (Figure 1). The experiments followed a schedule shown in Figure 1a. Two trametinib-resistant cell lines displaying either low (29_TRAR) or high (21_TRAR) ERK1/2 activity were investigated. While trametinib-resistant 29_TRAR cells did not maintain elevated levels of p-MEK1/2 detected in drug-naïve DMBC29 cells, trametinib withdrawal restored phosphorylation of MEK1/2 in 29_TRAR cells to the level found in DMBC29 cells (Figure 1b). Re-exposure of cells to trametinib resulted in a strong inhibition of MEK1/2 phosphorylation, which could be fully reversed by a second round of drug withdrawal. Although we have previously reported that 29_TRAR cells harbor de novo heterozygous mutations in *MAP2K2* giving rise to MEK2^L201V^ and MEK2^F57V^ variants [22], an almost undetectable level of phosphorylated MEK1/2 in resistant cells is obviously not due to genomic alterations but rather adaptive changes, as phosphorylation of MEK1/2 could be fully restored by trametinib withdrawal (Figure 1b). The enhanced phosphorylation of ERK1/2 was detected during the first and second round of drug holiday in comparison to its low level in resistant cells grown in the presence of trametinib (Figure 1b). Interestingly, a substantial change in ERK1/2 phosphorylation could be detected already one day after trametinib cessation and one day after trametinib rechallenge (Figure 1c) indicating the remarkable plasticity of resistant cells. In the trametinib-resistant 21_TRAR cell line displaying a high p-ERK1/2 level, changes induced by drug holiday/re-exposure to trametinib were far less pronounced than those observed in the 29_TRAR cell line (Figure 1d,e). These discrepancies might be due to differences in original phenotypes of drug-naïve cell lines and adaptive mechanisms employed during the development of resistance to trametinib (Figure 1f).

### 3.2. Levels of MCL-1 and Its Endogenous Inhibitors, BIM and NOXA, Are Modulated in Trametinib-Resistant Melanoma Cells during Alternating Periods of Trametinib Withdrawal and Rechallenge 

Whole-exome sequencing of trametinib-resistant cell lines revealed that none of the acquired mutations in any component of core apoptotic machinery could be classified as possibly damaging (Appendix A). As MCL-1 is crucial for melanoma survival [49], we investigated its expression in trametinib-resistant cells during alternating periods of trametinib withdrawal and rechallenge. Transcript and protein levels of MCL-1 were significantly increased in 29_TRAR cells during the first and second round of drug holiday, whereas during re-exposure to trametinib they returned to the levels detected in resistant cells (Figure 2a,b). In 21_TRAR cells that expressed MCL-1 protein at high level, trametinib withdrawal did not change MCL-1 levels so markedly (Figure 2a,b). Of note, the MCL-1 transcript level was already significantly higher in 21_TRAR cells than in 29_TRAR cells (Figure 2c). Bearing in mind that the stability of anti-apoptotic protein MCL-1 and its inhibitor BH3-only protein BIM can be influenced by p-ERK1/2-mediated phosphorylation [50,51], we analyzed possible associations between changes in ERK1/2 activity and alterations in the levels of MCL-1 and BIM. Indeed, the enhanced activity of ERK1/2 (Figure 1b) in 29_TRAR cells on drug holiday was accompanied by increased MCL-1 protein level (Figure 2b) and reduced BIM level (Figure 2d). In 21_TRAR cells, small changes in the already high p-ERK1/2 level (Figure 1d) were associated with only a minor increase in the high level of MCL-1 (Figure 2b) and a slight reduction in the low level of BIM (Figure 2d). The substantial changes in p-ERK1/2 and BIM levels in 29_TRAR cells could be detected already one day after trametinib withdrawal or one day after re-treatment of cells (Figure 1c and Figure 2d, lower panel). To summarize, ERK1/2-dependent stabilization of MCL-1 and degradation of BIM might be considered as enhancing the pro-survival capacity of 29_TRAR cells after trametinib withdrawal and maintaining high pro-survival status of 21_TRAR cells, grown with and without trametinib (Figure 2e). While BIM can bind and inhibit all pro-survival proteins, NOXA is the BH3-only protein capable of inhibiting mostly MCL-1 [52]. NOXA was significantly increased at the transcript level in trametinib-resistant cells during the first and the second round of drug holiday. Re-exposure to trametinib reduced NOXA transcript to the levels (in 21_TRAR cells) or below the levels (in 29_TRAR cells) detected prior to drug holiday (Figure 2f), which corresponded well with changes in NOXA protein levels (Figure 2d). Thus, NOXA protein enhanced by drug holiday might diminish the pro-survival status of melanoma cells. It should be noted, however, that the pro-survival potential of resistant melanoma cell lines used in this study was sufficient to prevent extensive cell death in response to either drug withdrawal or drug rechallenge [53]. Therefore, the next question was how resistant melanoma cells displaying different pro-survival capacity would respond to agents affecting the anti-apoptotic activity of MCL-1.

### 3.3. MCL-1 Inhibitor, S63845 Induces Apoptotic Signaling Assessed as an Accumulation of Annexin V-Positive Cells, Caspase-3/-7 Activation and Phosphorylation of Histone 2AX (γ-H2AX) with Efficiency Dependent on Melanoma Cell Phenotype

To challenge trametinib-resistant melanoma cells, we used S63845, a synthetic inhibitor of MCL-1 that has been designed to specifically bind to BH3-binding groove of MCL-1 [41]. We first examined changes induced by 1 μM S63845 in a total number of viable cells (Figure 3a) and the percentages of Annexin V-positive apoptotic cells (Figure 3b). ABT-263 and ABT-199, BH3-mimetics interacting with cell death-preventing proteins other than MCL-1 were used for comparison. S63845 reduced overall viability more efficiently than ABT-263 and ABT-199 (Figure 3a). Annexin V/PI staining revealed that 29_TRAR cells were more sensitive to S63845 than 21_TRAR cells (Figure 3b). Interestingly, S63845 was more efficient in trametinib-resistant melanoma cells on drug holiday than in trametinib-naïve melanoma cells (Figure 3b). EC_50_ values (Figure 3c) were in the nanomolar range of concentrations for 29_TRAR melanoma cells grown with trametinib (0.1 μM S63845) and without trametinib (0.7 μM S63845). For 21_TRAR cells grown in the presence/absence of trametinib and drug-naïve DMBC21 and DMBC29 cells, much higher S63845 concentration than 2 μM would be necessary to reach EC_50_, indicating that these cells were insensitive to S63845. S63845 at 1 μM increased the percentages of cells with active caspase-3/-7 to higher values in the 29_TRAR cell lines than in the 21_TRAR cell lines as assessed by time-lapse fluorescence microscopy (Figure 3d). This corresponds well with the extent of phosphatidylserine flipping to the outer side of the membrane (Figure 3b) indicating that accumulation of Annexin V-positive cells in response to 1 μM S63845 was preceded by caspase-3/-7 activation. Phosphorylation of histone H2AX on serine 139 (γ-H2AX), which was upregulated in those S63845-treated melanoma cell populations that displayed high percentages of Annexin V-positive cells suggests that apoptosis induced by the MCL-1 inhibitor was accompanied with the DNA damage (Figure 3e).

### 3.4. S63845 Significantly Reduces the Level of NOXA Protein

Treatment of resistant melanoma cells and their drug-naïve counterparts with 1 μM S63845 slightly increased MCL-1 protein levels (Figure 4a) without affecting MCL-1 mRNA levels (Figure 4b, left panel). This is consistent with previously published results for colon carcinoma cell lines showing that S63845 by binding to MCL-1 inhibits its activity but also increases MCL-1 protein half-life [41]. S63845 did not markedly alter the NOXA transcript level (Figure 4b, right panel) but significantly reduced the NOXA protein level (Figure 4c). This is the first report showing that S63845 can reduce the NOXA protein level in cancer cells. No substantial effects of S63845, ABT-263 and ABT-199 were detected in activities of MEK1/2 and ERK1/2 in trametinib-resistant melanoma cells (Figure 4d).

## 4. Discussion

Acquired resistance to targeted therapies is a serious clinical problem [18,54,55]. Patients resistant to such a treatment have limited therapeutic options, partially due to incomplete understanding of melanoma heterogeneity and plasticity associated with epigenetic reprogramming. Therefore, continued research is needed to overcome or suppress resistance to BRAF^V600^/MEK1/2 inhibitors, and both the rechallenge with BRAF and MEK inhibitors and novel drug targets are considered as potentially effective treatment modalities [8,12,56,57]. Our previous study revealed that the pattern of genetic/non-genetic resistance-associated alterations was distinct for each of eleven melanoma resistant cell lines [22,44]. Trametinib-resistant melanoma cell lines that have been chosen for the present study displayed different activity of MEK1/2 and ERK1/2. This study revealed that p-MEK1/2^high^/p-ERK1/2^high^ cell line (21_TRAR) was less responsive to trametinib withdrawal/rechallenge than p-MEK1/2^low^/p-ERK1/2^low^ cell line (29_TRAR). The amplitude and duration of ERK1/2 signaling are diverse in different biological contexts, which can determine various outputs including proliferation, growth inhibitory signaling, cell death, and antitumor immunity [58,59,60,61,62,63,64]. We investigated the association between drug holiday/rechallenge-induced changes in ERK1/2 phosphorylation and the levels of the components of the apoptotic machinery, anti-apoptotic MCL-1, and its endogenous inhibitors, BIM and NOXA. Several complex relationships between these proteins that have been shown previously [50,51,65,66,67,68] are reflected in our study. MCL-1 protein can be stabilized directly by ERK1/2-mediated phosphorylation at Thr 163 in the PEST region [50]. In addition, enhanced phosphorylation of ERK1/2 has been associated with elevated MCL-1 transcript level [66]. Phosphorylation of pro-apoptotic BIM by ERK1/2 has been shown as causing its dissociation from the complex with MCL-1 and proteasomal degradation [51,67,68]. Therefore, the hyperactivation of the BRAF/MEK/ERK pathway in many cancer types may suppress BIM protein level and evade apoptosis [68], whereas therapies targeting this pathway have been recognized as leading to the accumulation of BIM and induction of apoptosis [45,69,70,71]. We extended this view by showing that drug holiday-enhanced activity of ERK1/2 that was associated with the reduction in BIM level might be partially compensated by an increase in the NOXA level, a more specific endogenous inhibitor of MCL-1. It has been shown that while the introduction of BRAF^V600^ into melanocytes increased NOXA levels [72], inhibitors of BRAF or EGFR depleted NOXA protein [70] indicating that the NOXA level may also be dependent on BRAF/MEK/ERK signaling. Trametinib-reduced NOXA level has been recently reported in drug-naïve rhabdomyosarcoma cells [73]. NOXA has a high affinity for MCL-1 protein and plays a key role in MCL-1 localization and stability [74,75,76,77]. The balance between MCL-1 and NOXA may determine the susceptibility of cancer cells to drug-induced apoptosis [78], and the degradation of NOXA/MCL-1 complexes has been shown as determining the response of cancer cells to antimitotic treatment [79].

Phenotypic plasticity as a mechanism of melanoma cell resistance to therapy is a well-recognized clinical problem [21,80,81]. Several reports have demonstrated the role of modulation of pro- and anti-apoptotic proteins in the regulation of cell response to targeted therapies, and these studies have been intensified when targeted therapeutics were found to exert a low potency in inducing cancer cell death [82,83,84,85,86]. BH3-mimetics, ABT-199 (Venetoclax), a selective inhibitor of BCL-2, and ABT-263 (Navitoclax), an inhibitor of BCL-2, BCL-XL and BCL-w have been developed for clinical use, however, they have exerted low efficacy in most malignancies, except those of hematological origin [87]. Navitoclax in combination with dabrafenib and trametinib is still a subject of active study, not recruiting phase I/II clinical trial in patients with *BRAF*-mutant melanomas (NCT01989585). Trametinib-resistant melanoma cells grown with or without trametinib did not respond with substantial apoptotic death to ABT-263 or ABT-199 in our study indicating that these cells did not rely on anti-apoptotic proteins BCL-2, BCL-XL, and/or BCL-w. Much higher efficacy in inducing apoptosis in trametinib-resistant melanoma cell lines was obtained with S63845. This inhibitor designed to bind to the BH3-binding groove of MCL-1 has shown a high efficacy (IC_50_ < 1 μM) in vivo in mouse models of hematological malignancies [41]. It has been shown that MCL-1 and MEK protein levels may determine antileukemic treatment response to S63845 and trametinib, used alone or in combination [88]. Most solid tumor-derived cell lines exerted low sensitivity to S63845, and only 25% (three out of 12) melanoma cell lines were found sensitive to this agent [41]. Drug-naïve melanoma cell lines used in our previous study were relatively resistant to S63845, however, they were more primed to undergo apoptosis than melanocytes as shown using whole-cell BH3 profiling [45]. S63845 when used in combination with ABT-199 was more potent in melanoma cells without *BRAF* mutation [89] whereas combined with encorafenib, a BRAF^V600^ inhibitor, induced apoptosis in significantly higher percentages of drug-naïve BRAF^V600^ melanoma cells than either drug alone [45]. In another study, it has been demonstrated that trametinib was synthetic lethal with the MCL-1 inhibitor AZD5991 by promoting apoptosis of melanoma cells and inhibiting the growth of patient-derived xenografts [90]. There are no reports on using S63845 in melanoma cells resistant to targeted therapeutics and after cessation of treatment (drug holiday). The important finding of our study is that the pro-apoptotic activity of S63845 in trametinib-resistant melanoma cells is higher than in their drug-naïve counterparts. This means that the treatment with MCL-1 inhibitor might be more beneficial for melanoma patients who developed resistance to MEK inhibitors than for patients before the treatment with MEK inhibitors. Of note, while this was observed in 29_TRAR cells regardless of the presence/absence of trametinib, in 21_TRAR cells such an effect was obtained only on drug holiday. The level of MCL-1 protein might be one of the factors underlying the differences in the response to S63845. While 29_TRAR cell line (MCL-1^low^) displayed the highest sensitivity to S63845, trametinib withdrawal increased MCL-1 level and reduced sensitivity of these cells to S63845. Low sensitivity of 21_TRAR cell line to S63845 in the presence of trametinib was increased on drug holiday suggesting that additional factors may impact S63845 efficacy in these permanently MCL-1^high^ melanoma cells. Multiple changeable factors to account for MCL-1 level and S63845 activity should be considered. Our findings that (1) drug holiday induced phosphorylation of ERK1/2, which is presumably responsible for the enhancement of MCL-1 stability and reduction in the BIM stability; (2) drug holiday increased NOXA level possibly diminishing the MCL-1 availability, suggest that alterations in the functional interactions among the BCL-2 family members might influence the susceptibility of trametinib-resistant melanoma cells to S63845. Surprisingly, we found that S63845, a selective inhibitor of MCL-1, could also induce NOXA protein depletion. The reduction in NOXA level was detected uniformly in trametinib-resistant cell lines grown with or without trametinib and drug-naïve melanoma cells. While S63845 has been recognized as promoting the MCL-1 protein stabilization [41], presumably by interfering with the interaction between MCL-1 and MULE (MCL-1 ubiquitin ligase E3) [91], NOXA depletion induced by S63845 has not been considered as contributing to MCL-1 stabilization. This view is supported by a report showing that MCL-1 can stabilize NOXA protein by interaction with its BH3 domain, and the disruption of MCL-1:NOXA complexes can lead to proteasomal degradation of NOXA [92]. Therefore, it is reasonable to suggest that S63845, by binding the region of MCL-1 normally occupied by NOXA, can prevent the formation of MCL-1:NOXA complexes leading to NOXA degradation and stabilization of MCL-1. Various factors influencing the NOXA level in solid cancer have been identified [93,94], however, the impact of MCL-1 synthetic inhibitor has not been reported.

Considering the potential therapeutic applications of MCL-1 inhibitors, a safe therapeutic window in cancer patients is suggested to be achievable [41,95], regardless of the pleiotropic functions of MCL-1 [96,97]. S63845-related MCL-1 inhibitors S64315/MIK665 are currently evaluated in clinical trials (ClinicalTrials.gov ID: NCT04702425; NCT04629443; NCT03672695). Therefore, it would be interesting to verify our findings in clinics for the therapeutic benefit of melanoma patients that developed resistance to targeted therapy.

## 5. Conclusions

This study, although preclinical, provides several insights into the phenotypic plasticity of trametinib-resistant melanoma cells that might be considered in future clinical studies to enable the development of more personalized treatment for patients with melanoma resistant to BRAF^V600^/MEK inhibitors.

As high activity of ERK1/2 may increase the level of NOXA and stability of MCL-1 and reduce the stability of BIM, differences in the level of ERK1/2 activity between different melanomas resistant to trametinib as well as alterations in ERK1/2 activity induced by the termination of treatment might impact the pro-survival status of melanoma cells and outcome of the pharmacological inhibition of MCL-1. S63845, designed to specifically inhibit MCL-1 activity by binding to its BH3-binding groove, was also capable to induce the depletion of NOXA protein, an endogenous inhibitor of MCL-1. Thus, our study supports the notion that it is necessary to assess the main molecular determinants of activity of novel therapeutic target to define requirements for a highly efficacious new treatment regimen. It should be preferentially performed on resistant cells in the presence of a drug that caused resistance but also after drug cessation. Our findings also suggest that the treatment with MCL-1 inhibitors might be more beneficial for melanoma patients who developed resistance to MEK inhibitors than patients before this treatment.

## Figures and Tables

**Figure 1 cancers-15-04799-f001:**
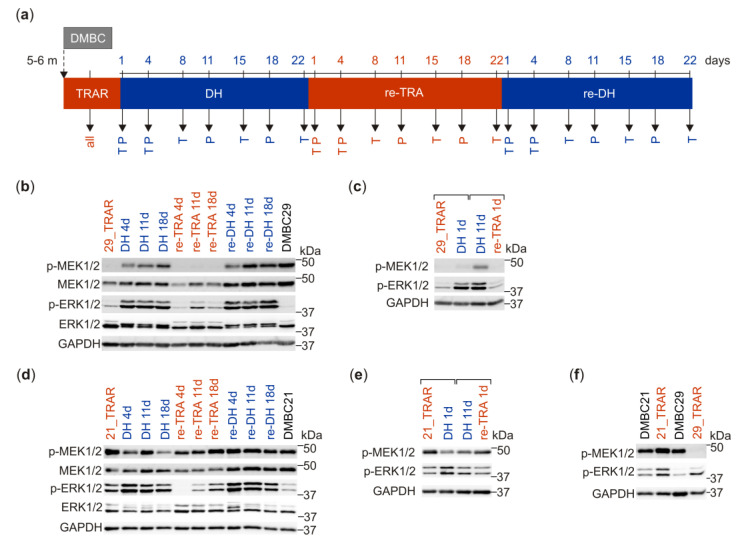
The BRAF/MEK/ERK pathway is differently affected by drug holiday and drug re-exposure in trametinib-resistant melanoma cells displaying either p-ERK1/2^low^ (29_TRAR) or p-ERK1/2^high^ (21_TRAR) phenotype. (**a**) The outline of experiments. Stable resistant cell lines (29_TRAR and 21_TRAR) were obtained from two patient-derived drug-naïve cell lines (DMBC29 and DMBC21, respectively) after continuous exposure to increasing concentrations of trametinib for 5–6 months (5–6 m). Trametinib-resistant melanoma cells (TRAR) were subjected to alternating periods of trametinib withdrawal (blue) and rechallenge (red); DH, drug holiday; re-TRA, re-exposure to trametinib after DH; re-DH, the second round of DH. Biological material collected at indicated days was subjected to assessment of gene expression at the transcript (T) and protein (P) levels. Resistant cells grown in the presence of trametinib were assessed in each experiment as a reference. (**b**,**d**) Cell lysates were collected from TRAR cells grown with (red) or without (blue) trametinib for indicated time and immunoblotted with specified antibodies. Western blots are representative of three independent experiments. (**c**,**e**) Cell lysates were collected from TRAR cells grown in the presence of trametinib, one day after trametinib withdrawal (DH 1d), on a drug holiday for eleven days (DH 11d), and one day after trametinib rechallenge (re-TRA 1d). Western blots are representative of two independent experiments. (**f**) Comparison of MEK1/2 and ERK1/2 activities in trametinib-resistant melanoma cell lines and their drug-naïve counterparts by immunoblotting. Full Western Blot images can be found in Appendix A.

**Figure 2 cancers-15-04799-f002:**
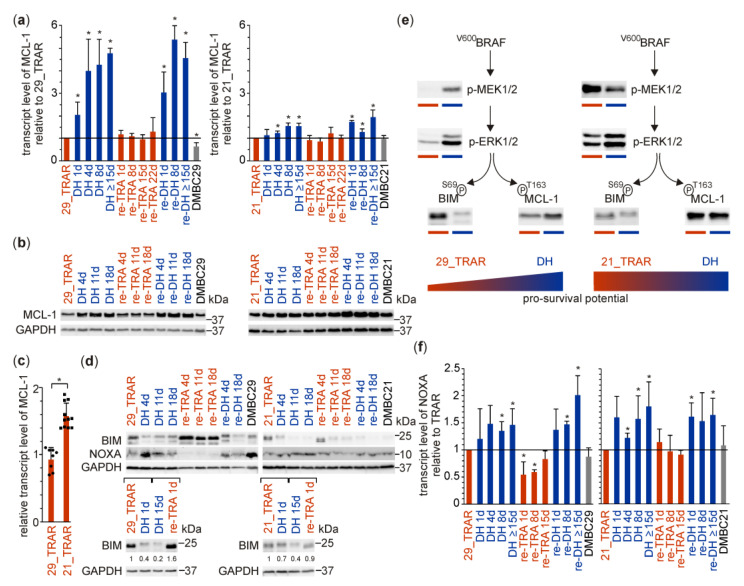
Levels of pro-survival MCL-1 and its endogenous inhibitors pro-apoptotic BIM and NOXA are affected by drug holiday/rechallenge in trametinib-resistant melanoma cells. Trametinib-resistant melanoma cells (TRAR) were subjected to alternating periods of trametinib withdrawal (drug holiday: DH and re-DH) and rechallenge (re-TRA). (**a**) Transcript levels were assessed by qRT-PCR. Fold change in MCL-1 transcript levels in DH, re-TRA, and re-DH cells relative to the MCL-1 transcript level in TRAR cells, normalized to RPS17 as a reference gene. Mean values ± S.D. of *n* = 3–5 biological replicates, ** p* < 0.05. (**b**) MCL-1 protein levels were assessed by immunoblotting. Western blots are representative of three independent experiments. (**c**) The relative MCL-1 transcript level, 21_TRAR cells vs. 29_TRAR cells assessed by qRT-PCR. Mean ± S.D., 19 independent biological samples, displayed as overlapping black dots, were used: *n* = 8 samples per group of 29_TRAR and *n* = 11 samples per group of 21_TRAR, ** p* < 0.05. (**d**) Analysis of changes in BIM and NOXA protein levels by immunoblotting. Western blots are representative of two independent experiments. Optical density quantification of changes in the BIM level one day after drug withdrawal/rechallenge normalized to GAPDH level is shown below the blots. (**e**) Scheme summarizing the potential relationship between the activity of ERK1/2 and the levels of BCL-2 proteins: anti-apoptotic MCL-1 and its endogenous inhibitor pro-apoptotic BIM. In 29_TRAR (p-ERK1/2^low^/BIM^high^/MCL-1^low^), the large enhancement of ERK1/2 activity during drug holiday is associated with increased MCL-1 level and reduced BIM level. In 21_TRAR cell line showing high activity of ERK1/2, drug holiday only slightly increases p-ERK1/2 level that is associated with a minor reduction in the low level of BIM, whereas the high MCL-1 level remains almost unchanged. (**f**) Analysis of changes in NOXA transcript levels during alternating periods of trametinib withdrawal and rechallenge. Mean ± S.D. of *n* = 3–5 biological replicates, ** p* < 0.05. Full Western Blot images can be found in Appendix A.

**Figure 3 cancers-15-04799-f003:**
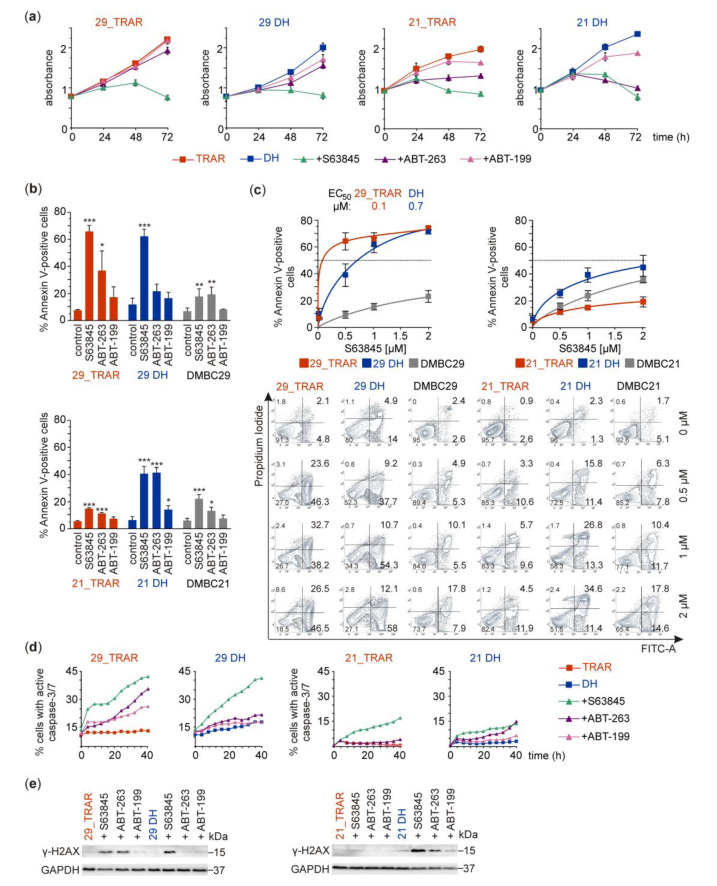
Pro-apoptotic effects of S63845 in trametinib-resistant melanoma cells grown continuously with trametinib (TRAR) or without trametinib (DH) for 8 days. (**a**) The overall viability of melanoma cells after treatment with BH3 mimetics S63845, ABT-263, or ABT-199 at 1 μM was determined with APA assay. The mean values ± S.D. of one representative experiment performed in triplicates. (**b**) Melanoma cells were exposed to 1 μM S63845, ABT-263, and ABT-199 for 44 h. Percentages of apoptotic cells was displayed as the sum of PI/Annexin V double positive and Annexin V-positive staining. Mean ± S.D. of *n* = 3–4 biological replicates. ** p* < 0.05, *** p* < 0.01, **** p* < 0.001. (**c**) Concentration-response curves were generated using GraphPad Prism 9.3. The EC_50_ values, the concentrations of S63845 at which cytotoxicity measured as Annexin-V-positive cell fraction reached 50%, were calculated only for 29_TRAR and 29 DH cell populations. Representative contour plots showing fractions of Annexin V-positive cells and/or PI-positive cells are included. (**d**) BH3 mimetics were used at 1 μM in resistant cells grown with (TRAR) or without trametinib (DH) to assess caspase-3/7 activation that was monitored by time-lapse fluorescence microscopy (IncuCyte ZOOM). Results of a representative experiment performed in triplicates are shown; *n* = 2. (**e**) Melanoma cells were exposed to 1 μM S63845, ABT-263, and ABT-199 for 24 h to assess drug-induced changes in the levels of γ-H2AX. Western blots are representative of two independent experiments. Full Western Blot images can be found in Appendix A.

**Figure 4 cancers-15-04799-f004:**
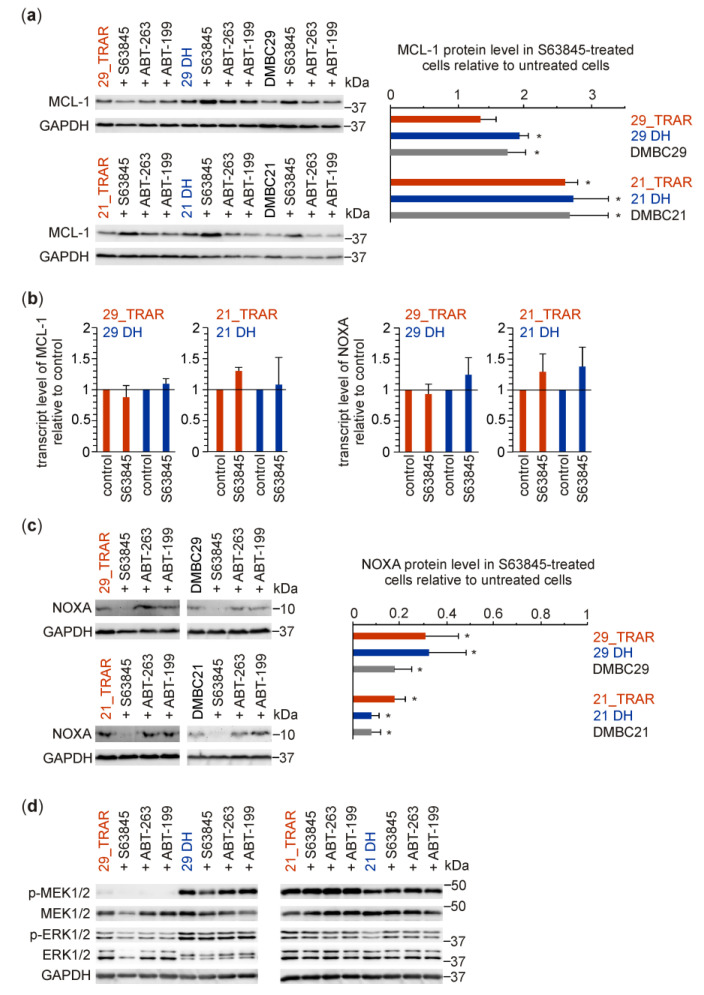
S63845-mediated inhibition of MCL-1 activity is accompanied by increased MCL-1 protein level and reduced NOXA protein level without changes in transcript levels of MCL-1 and NOXA and activity of MAPK/ERK pathway in trametinib-resistant melanoma cells grown continuously with trametinib (TRAR) or without trametinib (DH) for 8 days. (**a**) Western blot analysis of MCL-1 protein level in melanoma cells treated with S63845, ABT-263, and ABT-199 at 1 μM for 25 h. Right panel: optical density quantification of the MCL-1 level normalized to GAPDH level and represented as fold-change in S63845-treated cells compared to untreated cells. * *p* < 0.05. (**b**) The relative (S63845-treated vs. untreated) transcript levels of MCL-1 and NOXA were assessed by qRT-PCR. Mean ± S.D. *n* = 3. (**c**) Effects of 1 μM S63845, ABT-263, or ABT-199 on NOXA protein levels. Left panel: representative images from immunoblotting of cell lysates from trametinib-resistant and drug-naïve melanoma cells untreated and treated with 1 μM BH3-only mimetics. Right panel: optical density quantification of the NOXA level normalized to GAPDH level and represented as fold-change in S63845-treated cells compared to untreated cells. * *p* < 0.05. (**d**) Comparison of effects of BH3 mimetics (S63845, ABT-263, and ABT-199) used at 1 μM on MAPK/ERK signaling (p-MEK1/2, p-ERK1/2). Western blots are representative of two independent experiments. Full Western Blot images can be found in Appendix A.

## Data Availability

Raw sequencing data for DMBC21 and DMBC29 cell lines are available under accession numbers: E-MTAB-6978 (ArrayExpress) and ERP109743 (European Nucleotide Archive). Sequencing data for 21_TRAR and 29_TRAR cell lines are available under the numbers E-MTAB-7248 (ArrayExpress) and ERP111109 (European Nucleotide Archive). Other data presented in the manuscript are available on request from the corresponding author.

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
