# Peer review of "Pro-Apoptotic Activity of MCL-1 Inhibitor in Trametinib-Resistant Melanoma Cells Depends on Their Phenotypes and Is Modulated by Reversible Alterations Induced by Trametinib Withdrawal"

_cancers, 2023, doi:10.3390/cancers15194799_

Round 1

Reviewer 1 Report

In the manuscript, the author found that trametinib-resistant melanoma cells have distinct molecular phenotypes, and this phenotype is modulated by drug withdrawal. Further, the author correlated the pro-apoptotic and anti-apoptotic proteins with ERK activity in resistant cells. Using a specific inhibitor against MCL-1, they found a significant impact on trametinib-resistant cells. Interestingly, the MCL-1 inhibitor is not as effective for naïve melanoma cells.  The study was well conceived and performed and had a good experimental design. However, I believe there is room for improvement in the manuscript.

Major comment: -

1.       In most qPCR bar plots, the author plotted values relative to control with mean ± S.D. In the given representation, no error bar is shown in the control sample (for example, Fig. 2a and 2F plotted values for 29_TRAR and 21_TRAR). Therefore, I suggest the author present a bar graph showing individual biological values so readers can see the distribution of biological replicates around the mean. Such changes should be applied wherever it’s applicable.

2.       The author used a two-tailed Student’s t-test to analyze the statistical significance. However, most qPCR analyses compared the control vs. multiple groups in the same plot. The student T-test is not the ideal data analysis method in this circumstance. I suggest the author perform ANOVA analysis and re-analyze the qPCR data wherever applicable. The applied statistical method name should be written in the figure legend. 

3.       Fig.3a, the author claimed the S63845 reduced overall viability than ABT-263 and ABT-199. It is noticeable in 29_TRAR, but for other groups, it’s hard to conclude just by observing the mean value of the experiment performed in triplicates. The author should represent the graph with mean ± S.D. of the representative experiment, and the statistical test should be applied if possible.

4.       Fig.3C, the untreated control is missing in the contour plot; it is highly advised to add untreated control to see the overall change with treatment.

5.       Fig.3D, the authors should represent time-lapse fluorescence microscopy images for cells with active caspase 3/7.

6.       Fig.4, The author showed western blot calculation as a bar graph for different cell lines, including DMBC 29 and 21. However, the western blot image is missing for these cell lines in Fig. 4a and 4c.

7.       Fig 4a, MCL1 band intensity is noticeably lower in the S63485 treatment group compared to untreated 29_TRAR. If this is a representative western blot, I cannot understand how the author has more value for MCL1 protein levels in the bar graph for the same group. Although they mention comparing the band intensities of MCL protein with GAPDH, I suggest that the author provide a clear justification for quantifying their western blot.  

Minor comment: -

1.       I suggest providing the Ab catalog number and the primer sequence for the control gene RPS17.

2.       Fig 2C legend, author mentioned n=19. It is not clear to me what it means. Is it a biological replicate or a technical repeat? It should be clearly stated in the figure legend.

3.       In Fig2D, A better Western image should be provided (the NOXA blot for 21_TRAR).  

4.       Fig.3C, a contour plot should be in a better-quality image. Numbers in each quadrant are not perceivable.

5.       Fig. 3E, the author should describe a detailed legend at which time-point γH2AX was measured.

6.       21_TRAR melanoma cells treated with S36845 induces approx. 15% active caspase 3/7, and similar induction was observed in the 21_drug holiday group. But, in Fig. 3E, S36485 treated 21_TRAR is not induced γH2AX, whereas γH2AX is induced in the 21_drug holiday group. The result section should describe the discrepancy in results between γH2AX and caspase 3/7.

7.       The discussion session can be rewritten to make the manuscript more reader-friendly. I believe the author wrote detailed information, but it is not required in the discussion. Due to this, the reader may get lost while reading the discussion.  For example, line no. 410-414, the author wrote information for each inhibitor. I understand this inhibitor is used in this study. However, it is unnecessary to provide such information by citing old literature.

In conclusion, the author wrote their experimental observations instead of the concluding statement for the manuscript. I believe it should be rewritten 

Reviewer 2 Report

Hartman et al reported a phenotyping study on TRA resistant melanoma cell lines, which can be of clinically relevant. The manuscript can be improved in regard to the following points.

1. It is not clear how the pro-survival potential, as depicted in Fig.2(e), actually affects cell survival. Have authors measured cell viability and/or cell death on day 1, 4, 8, 15 of drug holiday or re-TRA?

2. In Fig.3(b), both S63845 and ABT-263 caused significant apoptosis in 29_TRAR cells . However, Fig.3(a) showed that viability of ABT-263 treated 29_TRAR cells was similar to control cells. Could authors comment on the disparity? Authors may want to perform a second viability assay, considering that the specific acid phosphatase activity as per cell may change across treatments.

3. The expression of NOXA in both 29_TRAR and 21_TRAR cells increased during drug holiday and returned to baseline during re-TAR, see Fig.2(d)(f). Does this suggest that NOXA expression is more dependent on TRA treatment itself?

4. The flow cytometry in Fig.3 should include a vehicle control (i.e. 0µM S63845).

5. The current introduction part looks like a detailed abstract and lacks a more general background.

6. The current conclusion part can be made more concise.

Please proofread the manuscript carefully for typos.

Reviewer 3 Report

In this article, Hartman et al. investigated trametinib withdrawal/rechallenge and MCL-1 inhibition in trametinib-resistant melanoma models with varying p-ERK1/2 levels. The results reveal reversible changes in ERK1/2 activity affecting pro-survival and pro-apoptotic protein balance. Additionally, the MCL-1 inhibitor S63845 showed apoptosis induction, especially in trametinib-resistant melanoma cells with specific p-ERK1/2/MCL-1/BIM/NOXA profiles, highlighting the importance of understanding cell phenotypes for effective therapy stratification. Overall rigor of the experiments and quality of the data are pretty high. I only have a few minor comments.

1.     The original 21_TRAC and 29_TRAC resistance lines were generated from 2019 or earlier according to the previous publication. Is there data/evidence to show that the resistance levels sustained in these two lines over the years?

2.     Only one concentration of trametinib was used for this study (50 nM). Could the effect on BRAF/MEK/ERK pathway described in Figure 1 dose-dependent?

3.     What is the baseline BRAF/MEK/ERK pathway key component activity level in 21_TRAC and 29_TRAC lines with complete DH or complete TRAC over 22 days?

4.     The gating in Figure 3c cuts through potentially Annexin/PI-positive populations. I would recommend using single-color controls and FMOs to determine better gating strategy and hence get more accurate data.

5.     The findings of the study are limited within the experiments done with the cell lines. It would be interesting to see whether it stands true in clinical setting using public database or clinical samples.

Round 2

Reviewer 1 Report

In the revised manuscript, the author did a great job and answered all the major and minor concerns in the previous version. The author provided sufficient explanation for major concerns wherever applicable. In addition, they made significant changes in the revised version as per requirement. I have a minor suggestion to the author that will help the manuscript be more reader-friendly. The calculation method used for qPCR analysis should be added to the supplementary information. 

Author Response

The calculation method used for qPCR analysis was added to the Supplementary information as requested.

Again, we would like to thank the Reviewer for constructive comments and several suggestions how to improve the manuscript. We are grateful for very positive comments about the revised version of the manuscript.  

Reviewer 2 Report

All my points have been well addressed. Thanks for authors' work.

Author Response

We are grateful for the positive comment about the revised version of the manuscript and our work done to improve it.